# Characterization of Resistance and Virulence of *Pasteurella multocida* Isolated from Pet Cats in South China

**DOI:** 10.3390/antibiotics11101387

**Published:** 2022-10-11

**Authors:** Haoyi Lin, Zhihui Liu, Yingchun Zhou, Weiguo Lu, Qian Xu

**Affiliations:** 1First Clinical Medical College, Guangzhou University of Chinese Medicine, Guangzhou 510006, China; 2Department of Clinical Laboratory, First Affiliated Hospital of Guangzhou University of Chinese Medicine, No. 16, Airport Road, Baiyun District, Guangzhou 510405, China

**Keywords:** *Pasteurella multocida*, whole-genome sequencing, resistance genes, virulence genes

## Abstract

*Pasteurella multocida*, one of the major zoonotic pathogens, may cause localized cellulitis, bacteremia, septic shock, and other symptoms in people. Accidental bites or scratches in close contact between humans and cats are one of the causes of *P. multocida* infection in humans. The prevalence of diseases due to *P. multocida* in humans may be rising as a result of the popularity of cat cafes in China. However, the epidemiology of *P. multocida* in cat-themed cafes in South China is yet to be reported. *P. multocida* in 92 healthy cats from four cafes with pet-cat themes was examined in their tonsils and oral cavities. The antibiotic susceptibility of each isolate was evaluated by using the disk-susceptibility testing method. The 19 *P. multocida* were selected for whole-genome sequencing (WGS), together with the genome data of eight previously described clinical strains isolated from humans, and the analysis of antimicrobial resistance (AMR) determinants, capsular genotyping, MLST genotyping, and virulence gene profiles was carried out. The results showed that 48.91% of cats (45/90) were positive for *P. multocida*. All isolated *P. multocida* stains were highly resistant to *erythromycin* (15 μg) (100%) and nonsensitive to *fluoroquinolones* (5 μg) (37.38%), while they were all susceptible to *penicillin* (10 untis), *tetracycline* (30 μg), *doxycycline* (30 μg), and *chloramphenicol* (30 μg). According to WGS analysis, those with the gyrA resistance gene were all insensitive to fluoroquinolone antibiotics. Virulence gene profiles showed that the genes *pfhA*, *nanH*, and *plpB* were practically all present in cats compared to humans. To conclude, increased antibiotic resistance, along with virulence factors, complicates the treatment of *P. multocida* infection. Thus, clinical treatment for *P. multocida* infection should be performed more cautiously.

## 1. Introduction

As a Gram-negative and facultatively anaerobic coccobacillus, *Pasteurella multocida* is one of the main zoonotic pathogens that can harm humans and livestock [1]. In animals, *P. multocida* is linked to chronic and acute diseases, including poultry cholera [2], swine atrophic rhinitis and suppurative bronchopneumonia [3], bovine respiratory disease [4], and respiratory disorders in varied hosts. However, many pets, such as dogs and cats, have *P. multocida* in their natural microflora [5]. Some research has shown that *P. multocida* is a proportion of the normal flora in the upper respiratory tract and nasopharynx of cats [6]. Therefore, cats are more commonly considered as healthy hosts of *P. multocida* [7,8]. *P. multocida* may be spread when a cat bites, scratches, or even licks an open wound on a person’s skin. In humans, infection occurs more frequently in those who are immunosuppressed or have major co-morbidities [9], but it has also been reported in healthy individuals [10,11]. In recent years, the rapidly increasing number of pet-cat-themed cafes may have fueled the problem of *P. multocida*-related infections as intimate contact with cats has been a major factor in the case reports of *P. multocida* infection in humans. *P. multocida* infection has been linked to localized cellulitis; bacteremia [12]; and, on rare occasions, even septic shock and empyema [13]. A few cases of peritonitis linked to peritoneal dialysis have been reported [14,15,16].

Much of the current *P. multocida* study has concentrated on livestock and its virulence. However, using broad-spectrum antibiotics can contribute to the formation of multi-drug resistant bacteria. Moreover, fewer scientists are investigating the *P. multocida* resistance genes in pets. During our study, we reported the epidemiology of *P. multocida* in cat-themed cafes in south China with the purpose of evaluating the occurrence of *P. multocida*, identifying antimicrobial susceptibility profiles and resistance genes, and revealing virulence gene patterns in *P. multocida*.

## 2. Results

A total of 242 strains were obtained from 93 cats in the themed cafes for our study. Among these cats, we found that 48.38% were positive for *P. multocida* isolation (45/93). Additionally, the *P. multocida* took the largest percentage of the 242 isolates, followed by *Neisseria zoodegmatis* and *Staphylococcus felis* (Figure 1), suggesting that the flora of *P. multocida* are common in cats. The antibiotic susceptibility of all identified *P. multocida* was investigated by using agar disk diffusion tests, as recommended by document (M45 3rd edition, 2016) [17] CLSI. All 45 *P. multocida* isolates were shown to be resistant to *erythromycin* (15 μg) Erythromycin; this implied that *erythromycin* (15 μg) Erythromycin had an insignificant effect on these *P. multocida* stains. The susceptibility rates for other antimicrobial agents such as *moxifloxacin* (5 μg), *levofloxacin* (5 μg), and *ceftriaxone* (30 μg) moxifloxacin, levofloxacin, and ceftriaxone were 62.22% (28/45), 62.22% (28/45), and 80.00% (36/45), respectively. Additionally, the results of the antimicrobial susceptibility tests showed that every isolate of *P. multocida* was sensitive to *penicillin* (10 untis), *tetracycline* (30 μg), *doxycycline* (30 μg), and *chloramphenicol* (30 μg) (Table 1).

For the purpose of better understanding the antimicrobial resistance genes, the phylogenetic relationship, and the virulence genes of *P. multocida* isolates, whole-genome sequencing data were used.

The antimicrobial resistance genes were examined through the CARD database in order to better comprehend the mechanisms underlying these *P. multocida* isolates’ antimicrobial resistance. The CARD data revealed that, with the exception of the resistance gene *gyrA*gyrA, all isolates carried the same antimicrobial resistance genes (Figure 2). All sequences contained the resistance genes *fabI*, *CRP*, *EF-Tu*, *rsmA*, *msbA*, *hmrM*, and *PBP3*, but only pm7, pm32, pm33, pm98, pm101, pm106, pm111, pm112, and pm113 strains contained the mutations in *gyrA*. The strains harboring the *gyrA* resistance gene were nonsusceptible to *levofloxacin* (5 μg) and *moxifloxacin* (5 μg), which was consistent with the results of the antimicrobial susceptibility testing.

On the basis of the MLST genotyping results, three *P. multocida* (pm18, 98, and 111) could not be classified by the MLST databases. In addition, there were six *P. multocida* (pm92, 94, 95, 99, 107, and 112) that could not be perfectly matched to the house-keeping gene *est*; therefore, their MLST typing indicated the closest ST264, and we referred to that as ST264*. Other MLST sequence types were diverse, comprising two ST3, three ST9, one ST11, one ST15, four ST25, one ST39, one ST54, one ST130, two ST155, one ST242, and two ST357. In order to find the correlation between *P. multocida* isolated from cats and humans, we gathered 8 whole-genome sequencing (WGS) data sets from *P. multocida* obtained from humans, which were found in GenBank (Accession numbers: CP026744, CP023516, CP023972, LS483473, NBTJ02000002, UGSW01000003, UGSV01000001, and UGSU01000002). The phylogenetic tree showed that pm98, together with UGSV01000001, was closely linked to strains obtained from humans. The results also revealed that the same ST typing was on the same branch (Figure 3, Image A).

The pathogenicity of *P. multocida* is often linked to a variety of virulence factors [18]. Additionally, the potential virulence genes in the *P. multocida* genome were found by using the PmGT virulence genotyping database. In the epidemiological research of *P. multocida*, the following 24 genes that encode virulence factors have usually been analyzed [19]. The genes *ptfA*, *fimA*, *hsf-1*, *hsf-2*, *pfhA*, *pfhB*, and *tadD* code for fimbriae and other adhesins. *ToxA* and *pmHAS* encode toxin and hyaluronidase, respectively. Both *exbB*, *exbD*, *tonB*, *hgbA*, *hgbB*, *fur*, and *tbpA* encode iron acquisition proteins. *SodA* and *sodC* encode superoxide dismutase, while *nanB* and *nanH* encode sialidases. Additionally, the genes *ompA*, *ompH*, *oma87*, and *plpB* encode the outer membrane proteins [19,20,21,22,23]. During our study, we examined the virulence genes of the 19 *P. multocida* acquired from cats and the eight *P. multocida* isolated from humans. As shown in Image B (Figure 3), we found cats almost universally possess the virulence genes *pfhA*, *nanH*, and *plpB*, in contrast to humans. Additionally, all cat isolates expressed the virulence gene *sodA*, except pm98. The heatmap also demonstrated that the same MLST type of *P. multocida* isolated from cats or people had the same virulence profile. Furthermore, the virulence profile of strain pm17 showed that it carried all 24 virulence genes.

## 3. Discussion

*Pasteurella multocida* is found naturally in cats’ oral cavities [6] and can transmit from cats to humans, resulting in infection; this phenomenon is not rare [12,13,14,15,16,24,25]. Due to the popularity of cat-themed cafes in China, close contact between customers and cats is going to promote the spread of *P. multocida* between these two groups, which may increase the frequency of human *P. multocida* infection cases. Further research is therefore required to analyze and explore these potentially infectious *P. multocida* strains.

The prevalence of *P. multocida* identified during our study (48.38%) was higher than that reported by Ferreira [7], who found 10.47% of positive samples in cats’ gingival mucosa (20/191). The positive rate of *P. multocida* isolated from the oral mucosa of cats, according to Freshwater, was 89.97% (368/409) [26], which was higher than our finding. The differences might be explained by the different living conditions of cats. During our study, the cats were derived from cats housed in coffee shops in a home environment comparable to Freshwater [26], under similar circumstances, and with similar separation rates. The variation in *P. multocida* separation rates might also be influenced by the cats’ own oral environment. The distribution of flora in the mouths of the cats might alter, depending on geography.

The virulence factors are important for *P. multocida* both in overcoming the host immune response and in boosting its survival in the host environment [27,28]. The *pfhA* gene, which encodes filamentous haemagglutinins, plays an important role in the early bacterial colonization of the upper respiratory tract [29]. Much research has shown that the *P. multocida* strains that were isolated from various hosts varied in their *pfhA* isolation rates. For example, compared to a far lower frequency of *P. multocida pfhA* isolation from cattle [29], the isolation rate of *P. multocida pfhA* from cats ranged from 0% to 23.8% [7,18,20]. Interestingly, we found that the *P. multocida* virulence gene *pfhA* was expressed in nearly all cats during the current study. The *nanH* gene is involved in epithelial surface colonization, similar to the function of *pfhA*, and its prevalence varies depending on host origin and geographical location [20,29]. During our study, the frequency of nanH was similar to *pfhA*, which may imply that they may act synergistically to promote bacterial colonization and survival in the oral cavity of cats. This could mean that people who visit these cat cafes and who get bitten by a cat are more likely to have *P. multocida* infection.

As for the treatment of *P. multocida*, antibiotics are currently recommended. Antibiotics, including *penicillin, fluoroquinolone*, and the second or third generation of *cephalosporin*, are commonly used to treat *P. multocida* infection [30]. Mu and Yang [14] dealt with the statistics and analysis of antibiotic-sensitive test results in the cases of the last three decades. Their research revealed that *P. multocida* had a very low rate of antibiotic resistance, and all the isolates were sensitive to *ceftriaxone* and *fluoroquinolone*. Based on our antibiotic-sensitive test results, the sensitive rates of *ceftriaxone* (30 μg) and *fluoroquinolones* (5 μg) were 80.00% and 62.22%, respectively. In particular, the resistance rate to *erythromycin* (15 μg) was found to be 100%. The data indicated an increasing prevalence of antibiotic-resistant *P. multocida*. As a result, the administration of antimicrobials for *P. multocida* infections should always be based on the results of antibiotic susceptibility testing.

In our study, we discovered that eight *fluoroquinolone*-resistant strains had mutations in the *gyrA D87G* gene. *Fluoroquinolone* prevents bacterial reproduction by inhibiting DNA gyrase (*gyrA* and *gyrB*) and topoisomerase IV [31]. Point mutations in the QRDR, which determines quinolone resistance, are closely linked to increased resistance to *fluoroquinolone*. A chromosomal point mutation in type IV topoisomerase (*parC* and *parE*) or DNA gyrase (*gyrA* and *gyrB*) is known as a QRDR mutation [32]. In our research, eight strains had an identical point mutation in *gyrA*. Additionally, *fluoroquinolone* insensitivity could also occur in strains without *gyrA* mutations. *HmrM*, for instance, belongs to the MATE family of multidrug efflux pumps. Based on earlier research, the *hmrM* gene majorly contributed to *norfloxacin* resistance in *H. influenzae* [33]. Thus, resistance to *fluoroquinolone* was considered to be multifactorial.

During our study, all isolates were resistant to *erythromycin* (15 μg). WGS analysis found that *CRP*, the regulator of the MdtEF-TolC efflux pump, was presented in all isolates. The MdtEF-TolC efflux pump is a member of the HAE-RND subfamily [34]. RND efflux pumps in Gram-negative bacteria can expel several kinds of antimicrobials from cells, which directly leads to multidrug resistance y [35]. It has also been discovered that MdtEF may extrude *erythromycin* under anaerobic conditions [34]. During our research, the whole-genome sequencing of *P. multocida* uncovered a variety of mechanisms that led to antibiotic resistance and demonstrated that *P. multocida* led to antibiotic resistance by modifying antibiotic target sites or by regulating the flow of antibiotics into or out of bacterial cells.

## 4. Materials and Methods

### 4.1. Sample Collection and Bacterial Isolate Identification

The samples were taken by using sterile swabs from the tonsils and oral cavities of 93 cats from four cat-themed cafes in different parts of Guangzhou, South China in 2022. All of the cafe cats in our study were in good health. They were energetic and had a voracious appetite. No respiratory or gastrointestinal illnesses existed, they had no fever, and they had not taken any antibiotics or other medications in the last three months. The cafes had enough room for them to live there. After taking the samples, the swabs were put in nutrient broth mediums and cultivated for 24 h at 35 °C in a 5 percent CO${}_{2}$ incubator. Then, broth medium was inoculated onto blood agar, which was incubated for 18 h. All isolates were directly identified by using matrix-assisted laser desorption ionization time-of-flight mass spectrometry (MALDI-TOF MS).

### 4.2. Antimicrobial Susceptibility Testing

Antimicrobial susceptibility was tested on all *P. multocida* strains that were collected. The disk-susceptibility testing method was used to determine the susceptibility profile as suggested by the Clinical and Laboratory Standards Institute (CLSI), and the breakpoint was interpreted according to the CLSI document (M45 3rd edition, 2016) [17]. The antimicrobial agents tested included *amoxicillin-cavulanate*(20/10 μg), *ampicillin* (10 μg), *penicillin* (10 units), *ceftriaxone* (30 μg), *moxifloxacin* (5 μg), *levofloxacin* (5 μg), *tetracycline* (30 μg), *doxycycline* (30 μg), *erythromycin* (15 μg), *azithromycin* (15 μg), *chloramphenicol* (30 μg), and *trimethoprim-sulfamethoxazole* (1.25/23.75 μg).

### 4.3. Whole-Genome Sequencing

A total of 19 *P. multocida* strains that showed antimicrobial nonsusceptibility to more than two antibiotic classes were chosen for whole-genome sequencing (WGS). The total genomic DNA of *P. multocida* was extracted by using a Steady Pure Bacterial Genomic DNA Extraction Kit (Accurate Biotechnology, Hunan, China) and stored at −20 °C until use. The Illumina Novaseq6000 platform at Sangon Biotech (Shanghai, China) was chosen for WGS with a 150 bp paired-end strategy. After assembling the reads, unqualified contigs covering fewer than 10 lengths or lengths lower than 500 bp were removed from further study.

### 4.4. Genomic Analysis

In order to predict open reading frames (ORFs) and related functional genes, we employed Rapid Annotation using Subsystem Technology (RAST) (https://rast.nmpdr.org/rast.cgi, accessed on 2 July 2022) to annotate the whole nucleotide sequence of *P. multocida*. We used PmGT (http://vetinfo.hzau.edu.cn/PmGT/home.php, accessed on 19 July 2022) to identify virulence factors encoding genes of *P. multocida* strains based on the WGS data [36]. Public databases for molecular typing and microbial genome diversity (https://pubmlst.org/bigsdb?db=pubmlst_pmultocida_seqdef&page=sequenceQuery, accessed on 25 July 2022) were used to determine multilocus sequence typing (MLST) from the assembled genome. We predicted resistomes from nucleotide data based on homology by using the resistance gene identifier (RGI) of the comprehensive antibiotic resistance database (CARD) (https://card.mcmaster.ca/analyze/rg, accessed on 25 July 2022). For the purpose of choosing antimicrobial-resistance genes from CARD, we set the threshold at > 60% identity of the matching region. On all of the chosen *P. multocida*, a phylogenetic tree analysis was carried out by using MEGA 11.0 and the NDtree database (https://cge.food.dtu.dk/services/NDtree/, accessed 19 July 2022) from the Center for Genomic Epidemiology [37,38,39]. Additionally, we performed heatmaps of the genes encoding virulence factors and antibiotic resistance by using TBtools (version 1.09876) [40].

## 5. Conclusions

In conclusion, we presented antimicrobial susceptibility profiles and resistance genes, as well as virulence genes, in *P. multocida* in cafe cats in south China. Our study demonstrated that *P. multocida* strains isolated from cafe cats were resistant and nonsusceptible to the commonly used antibiotics *erythromycin* (15 μg) and *fluoroquinolones* (5 μg), respectively. In addition, we speculated that the high expression of relevant virulence genes in *P. multocida* from cafe cats may make it easier for the species to colonize and survive in cells. Therefore, more caution should be used when administering clinical treatment for *P. multocida* infection. Additionally, the relevant health authorities should be concerned about the emergence of *P. multocida* strains that are nonsusceptible to antibiotics and that express a high level of various virulence factors, and solutions such as limiting antibiotic abuse should be suggested in response. As for consumers, close contact with cafe cats should be followed by washing hands thoroughly, and dining in the same area as the cafe cats should be avoided.

## Figures and Tables

**Figure 1 antibiotics-11-01387-f001:**
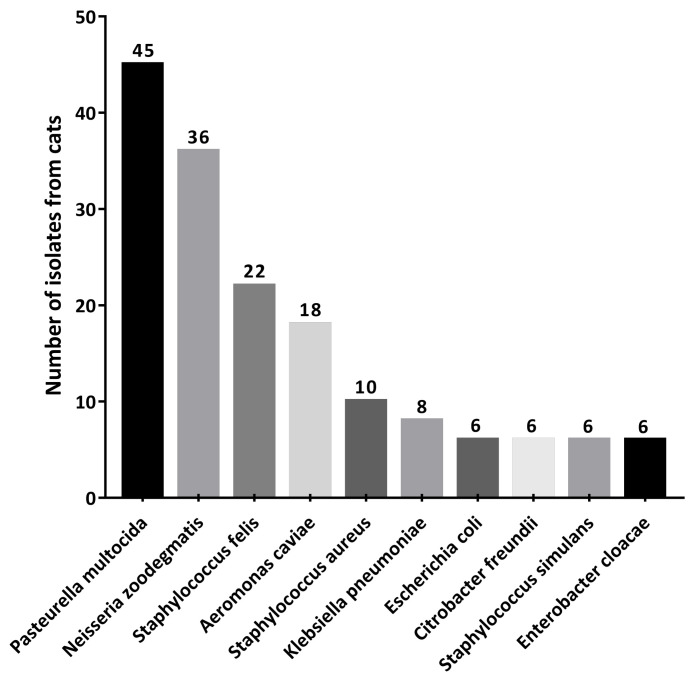
The top 10 strains from the 242 isolated strains from cafe cats. *P. multocida* had the largest percentage of the 242 isolates, followed by *Neisseria zoodegmatis* and *Staphylococcus felis*.

**Figure 2 antibiotics-11-01387-f002:**
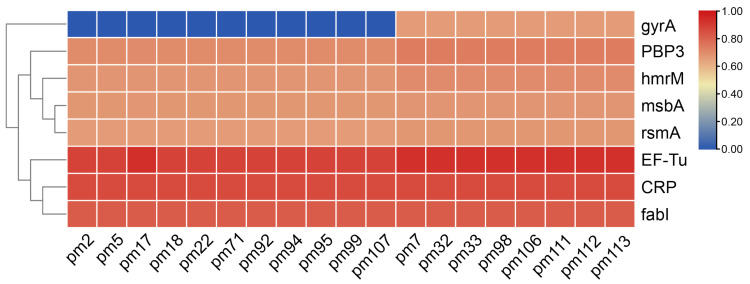
The heatmap of the genes associated with antibiotic resistance in *P. multocida*. The scale was determined by the identification of the matching region.

**Figure 3 antibiotics-11-01387-f003:**
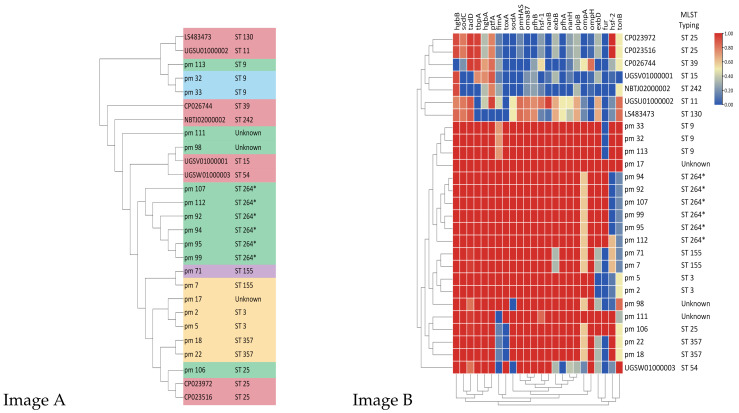
(**Image A**) Phylogenetic tree of 27 *P. multocida* strains. The eight *P. multocida* strains whose entire genomes were downloaded from GenBank are highlighted in red. In total, 19 *P. multocida* from cats were shown in other colors. Distinct colors in cat-derived *P. multocida* represented isolation from different cafes. ST264* indicated that their MLST type was the most similar to ST264. (**Image B**) The heatmap of the distribution of genes that encode the virulence factors of *P. multocida*. The relative abundance on a Z scale of 0 to 1 across each row served as the scale for the heatmap.

**Table 1 antibiotics-11-01387-t001:** Susceptibility rate of antimicrobials in *P. multocida* isolated from cafe cats.

Antimicrobial Class	Antimicrobial Agent	Disk Content	Number of Susceptible Strains (Susceptible Rate %)
Penicillins and β-lactam/β-lactamase inhibitor combinations	*Amoxicillin-clavulanate*	20/10 μg	44 (97.78)
	*Ampicillin*	10 μg	44 (97.78)
	*Penicillin*	10 units	45 (100.00)
Cephems	*Ceftriaxone*	30 μg	36 (80.00)
Fluoroquinolones	*Moxifloxacin*	5 μg	28 (62.22)
	*Levofloxacin*	5 μg	28 (62.22)
Tetracyclines	*Tetracycline*	30 μg	45 (100.00)
	*Doxycycline*	30 μg	45 (100.00)
Macrolides	*Erythromycin*	15 μg	0 (0.00)
	*Azithromycin*	15 μg	42 (93.33)
Others	*Chloramphenicol*	30 μg	45 (100.00)
	*Trimethoprim-sulfamethoxazole*	1.25/23.75 μg	40 (88.88)

## Data Availability

Not applicable.

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
