# Peer review of "Characterization of Resistance and Virulence of Pasteurella multocida Isolated from Pet Cats in South China"

_antibiotics, 2022, doi:10.3390/antibiotics11101387_

Round 1

Reviewer 1 Report

The manuscript entitled "Characterization of resistance and virulence of Pasteurella multocida isolated from pet cats in south Chin" is very well written and presented. The study of this pathogen in pets is not common, so the authors give a little regarding the zoonotic danger of this pathogen in cats, particularly in cat cafes where many people manipulate this animal. Overall is an excellent text that presents high novelty in this field.  

Some suggestion are:

Material and methods should be number 2 and not four. Methods should appear before results

Author Response

Response to Reviewer 1 Comments

Point 1: The manuscript entitled "Characterization of resistance and virulence of Pasteurella multocida isolated from pet cats in south Chin" is very well written and presented. The study of this pathogen in pets is not common, so the authors give a little regarding the zoonotic danger of this pathogen in cats, particularly in cat cafes where many people manipulate this animal. Overall is an excellent text that presents high novelty in this field. 

Some suggestion are:

Material and methods should be number 2 and not four. Methods should appear before results

Response 1:

Dear editors and reviewers:

Thank you very much for reviewing our manuscript. We also greatly appreciate your complimentary comments and suggestions.

We strongly agree with your suggestion, and this is how the article was initially written in “Introduction-Materials and Methods-Results-Discussion-Conclusion”. However, when the paper was submitted to Antibiotics, the template that was provided by Antibiotics stated that “Material and Methods” should be placed after “Results” and “Discussion” as the fourth major component of the paper. As a result, we did not change the order of “Materials and Methods” in the revised manuscript.

But your suggestion prompted us to revise some of the places listed as followed.

Line 53-54: “CLSI” was revised to “document (M45 3rd edition, 2016) CLSI“and reference 17 “CLSI. Methods for antimicrobial dilution and disk susceptibility testing of infrequently isolated or fastidious bacteria, 3rd ed. CLSI guideline M45. Clinical and Laboratory Standards Institute, Wayne, PA 2016.” was cited.

Line 83: WGS was changed to whole-genome sequencing (WGS) because WGS was used as an acronym without the complete name being mentioned in the previous parts.

I hope this revision can make my paper more acceptable. Thank you again for your help!

Best wishes,

Lin Haoyi.

Reviewer 2 Report

This manuscript presents data about characterization of resistance and virulence of Pasteurella multocida isolated from pet cats in south China. The topic of this article is of interest in the panorama of the One-health approach. However, I have some comments.

The genes name should be written in italic.

Please clarify the number of cats used to conduct the study.

Please clarify the terms antibiotics, drugs and antimicrobials in manuscript. Please always write in the same way and the correct term.

Results:

The sequence type (STs) identified by WGS are STs identified in Humans? Or in another  animal species.  Please clarify.

Discussion:

Line 129: Please change cafés for cafe.

Line: 139 – 140: The administration of antimicrobials should always be based on the results of antibiotic susceptibility testing. Please rewrite the sentence.  

Material and methods:

What year were the samples collected? Please clarify.

How do you know that cats are healthy ? Please clarify in the manuscript.

Had the animals’ taken antimicrobials before? How long ago? and which antimicrobials?

Please clarify the living conditions of cats.

Please clarify CLSI document.

Line: 180: Why the authors considered more than two antibiotic classes? Why not 3 antimicrobial classes?

Lines 176-178: You should write…Amoxicilin-clavunate (20/10 ug), Ampicilin (10 ug) …and so on.

 In table 1 please clarify the disk content units (g or ug?)

Author Response

Response to Reviewer 2 Comments

Dear editors and reviewers:

Thanks very much for taking your time to review this manuscript. I really appreciate all your comments and suggestions! Please find my itemized responses in below and my revisions and corrections in the re-submitted files.The main corrections in the paper and the responds to the reviewer’s comments are as flowing:

Point 1: The genes name should be written in italic.

Response 1: Thank you for pointing this out. Italics have been given to the gene names in the revised paper.

Point 2: Please clarify the number of cats used to conduct the study.

Response 2: The experiment involved 93 cats in all, as is stated in the RESULTS section. From 93 cats, we recovered a total of 242 strains, 45 of which were P. multocida and all originated from distinct cats.

Point 3: Please clarify the terms antibiotics, drugs and antimicrobials in manuscript. Please always write in the same way and the correct term.

Response 3: We are sorry for confusing the terms antibiotics, medicines, and antimicrobials in this article, and there is a distinction between them. The confusion has been cleared up as follows.

Line 18/143/145/147/150/175: we have corrected the “drug” into “antibiotic”.

Line 228: we have corrected the “antimicrobials” into “antibiotics”.

Point 4: Results:

The sequence type (STs) identified by WGS are STs identified in Humans? Or in another animal species. Please clarify.

Response 4: The sequence type (STs) identified by WGS are STs identified in another animal species. All STs were based on the database RIRDC MLST, and the seven house-keeping genes were adk, est, gdh, mdh, pgi, pmi and zwf.

Point 5: Discussion:

Line 129: Please change cafés for cafe.

Line: 139-140: The administration of antimicrobials should always be based on the results of antibiotic susceptibility testing. Please rewrite the sentence. 

Response 5: As suggested by the reviewer, we have corrected the “cafés” into “cafes” in Line 138. And the sentence “As a result, the treatment of P. multocida infection should be based on the results of drug susceptibility testing.” in Line 150-152 have changed into ”As a result, the administration of antimicrobials for P. multocida infection should always be based on the results of antibiotic susceptibility testing.”

Point 6: Material and methods:

What year were the samples collected? Please clarify.

How do you know that cats are healthy ? Please clarify in the manuscript.

Had the animals’ taken antimicrobials before? How long ago? and which antimicrobials?

Please clarify the living conditions of cats.

Please clarify CLSI document.

Response 6:

We collected the samples in 2022.

We added the health conditions, antibiotics usage and living conditions of cafe cats in the manuscript Line 181-184: And all of the cafe cats in our study were in good health. They were energetic and have a voracious appetite, no respiratory or gastrointestinal illnesses existed, they had no fever, and they hadn't taken any antibiotics or other medications in the last three months. The cafes had enough room for them to lived and played.

The CLSI document used in the manuscript was M45 3rd edition published in 2016, and the reference was clarified to “17. CLSI. Methods for antimicrobial dilution and disk susceptibility testing of infrequently isolated or fastidious bacteria, 3rd ed. CLSI guideline M45. Clinical and Laboratory Standards Institute, Wayne, PA 2016.” rather than “35. Clinical.; Institute, L.S. Methods for Antimicrobial Dilution and Disk Susceptibility Testing of Infrequently Isolated or Fastidious Bacteria. approved guideline CLSI document M45-A 2006”.

Point 7:

Line 180: Why the authors considered more than two antibiotic classes? Why not 3 antimicrobial classes?

Lines 176-178: You should write…Amoxicilin-clavunate (20/10 ug), Ampicilin (10 ug) …and so on

In table 1 please clarify the disk content units (g or ug?)

Response 7:

According to the findings of our tests for antimicrobial susceptibility, we discovered that the P. multocida resistance patterns isolated from cafe cats were primarily either fluoroquinolones and macrolides resistance or cephems and macrolides resistance. For whole genome sequencing and analysis, we, therefore, focused on the strains that are resistant to two or more antibiotics rather than resistant to three or more antibiotics classes.

We corrected the disk content units in table 1. Based on your comments, we had updated the manuscript to include the necessary antibiotic units. Thanks for your careful checks.

In all, we found the comments of yours are quite helpful, and we revised our paper point-by-point. Thank you and the review again for your help! And we hope the revised manuscript could be acceptable to you.

Bset wishes,

Lin Haoyi.

Reviewer 3 Report

Dear Editor,

The paper by Haoyi and all updated our knowledge about Pasteurella multocida genomics, virulence, and antibiotic resistance in cats.

The question is, are these strains isolated from animals the same as those isolated from patients after infection?

If so, is there a genetic or genomic distinction between the human and animal isolats?

Sincerely
